# Outcomes for university students following emergency care presentation for deliberate self-harm: a retrospective observational study of emergency departments in England for 2017/2018

Catherine Campbell, Joe Dodd, Igor Francetic 

Centre for Primary Care and Health Services Research, The University of Manchester, Manchester, UK

**Correspondence to**
Dr Igor Francetic;
igor.francetic@manchester.ac.uk

## ABSTRACT

**Objectives** Identify university-aged students and contrast their healthcare provision and outcomes with other patients in the same age group attending emergency departments for deliberate self-harm.

**Design** Retrospective cross-sectional observational study.

**Setting** Patients visiting 129 public hospital emergency departments across England between April 2017 and March 2018.

**Participants** 14 074 patients aged 18–23 visiting emergency departments for conditions linked to deliberate self-harm, 1016 of which were identified as university-aged students.

**Outcome measures** We study various outcomes across the entire patient pathway in the emergency department: waiting time to initial assessment on arrival at the emergency department, count of investigations delivered, discharge destination (patients refusing treatment or leave before being seen, referred to another provider or admitted to inpatient care, discharged with no follow-up) and unplanned follow-up visit within 7 days.

**Results** We find a statistically significant difference of 0.262 (−0.491 to −0.0327) less investigations delivered to students compared with non-students (about 8% compared with the baseline number of investigations for non-students). Stratified analyses reveal that this difference is concentrated among students visiting the emergency department outside of regular working hours (−0.485 (−0.850 to −0.120)) and students visiting for repeated deliberate self-harm episodes (−0.881 (−1.510 to −0.252)). Unplanned reattendance within 7 days is lower among students visiting emergency departments during out of hours (−0.0306 (−0.0576 to −0.00363)), while students arriving by ambulance are less likely to be referred to another provider (−0.0708 (−0.140 to −0.00182)) compared with non-students.

**Conclusions** We find evidence of less-intense investigations being delivered to patients aged 18–23 identified as students compared with non-students visiting emergency departments after an episode of deliberate self-harm. Given the high risk of suicide attempts after episodes of deliberate self-harm among students, our findings may highlight the need for more focused interventions on this group of patients.

## STRENGTHS AND LIMITATIONS OF THIS STUDY

⇒ We propose an approximate approach to identify students in administrative records for emergency departments (EDs) in England, focusing on the population of attendances between April 2017 and March 2018.

⇒ Supply-side confounders are accounted by a high-dimensional fixed effects, accounting for ED-specific combinations of day of the week and month of the year.

⇒ We study outcomes across the entire pathway for patients attending EDs after episodes of deliberate self-harm, comparing students and non-students.

⇒ The lack of a qualitative component in our study prevents a richer and more accurate interpretation of ED staffs' attitude towards self-harm patients.

⇒ Variability in the systems and protocol to assess and treat self-harm presentations may limit comparability across geographical areas despite we account for hospital-specific fixed effects to circumvent these differences.

## INTRODUCTION

Three-quarters of mental health disorders peak by the age of 25 years, an age-group associated with university attendance[1] and for which there is a large unmet need for mental healthcare.[2] As a result, there have been calls for increased research on student mental health.[3] Academic pressures and living far from social networks developed during early adolescence may pose additional stress on an already vulnerable group.[4] For instance, in the USA, among 274 surveyed universities, 88% of counselling centre's directors reported an increase in severe psychological problems over the previous 5 years.[5] However, in England, trends in incidence of severe mental health problems and suicide do not seem to be higher among students compared with the general population of the same age.[6–8] Nevertheless, from a service provision perspective, the

student population appears particularly vulnerable, especially in regards to limited availability and accessibility of mental health services for students.[9–11]

Our study relates to literature surrounding the quality of care within emergency departments (EDs) for deliberate self-harm (DSH) patients. First, ED staff attitude can exert a crucial effect on the future health-seeking behaviours of self-harm patients.[12] This suggests that if a patient experiences a negative reaction to seeking help for their ailment, they are less likely to seek similar help again. Moreover, there is evidence that patients who engage in self-harm or attempt suicide are more negatively evaluated and viewed as 'wasting resources' compared with patients attending for other reasons.[13] Timson *et al*[14] further find a correlation between staff knowledge of mental health treatment and negativity towards self-harm patients. Finally, there is ample evidence relating hospital attendance for non-fatal self-harm to subsequent risk of completed and attempted suicide, indicating that treatment for self-harm attendances is particularly sensitive.[15–18] These studies highlight the potential impact of staff attitudes and perception of mental health on care outcomes for students attending ED for DSH.

This paper builds on a growing body of literature regarding student mental health at university. Bolotnyy *et al*[19] find that economics PhD students at top-ranked American universities have experienced depressive symptoms at a rate of over two times the population average, with only a small percentage receiving help. Levels of depressive symptoms are even higher among graduate students attending European universities.[20] These studies demonstrate the prevalence and severity of mental health in universities, common across countries. As mentioned, demand-side barriers such as misperceptions about care-seeking may contribute to the treatment gap among university students.[21 22] In light of this, Acampora *et al*[23] find that among individuals who receive information about the benefits of seeking help, women present a higher likelihood of seeking help, resulting in improved mental health scores at follow-up. Holt and Powell[24] further find that 80% of university student attendees are not registered with a General Practitioner (GP). This may reflect in an apparent misuse of emergency care services, which has been attributed to students failing to seek healthcare on campus due to fear of vulnerability and breached confidentiality.[25]

Building on these stylised facts, we aim to explicitly study the healthcare provision and outcomes of university-aged students attending EDs for DSH, contrasting them with other DSH patients. In doing so, given that administrative hospital records do not include information about student status, we propose a proxy indicator to identify students in Hospital Episode Statistics (HES) records.

## METHODS
### Data and sample
Our study builds on the attendance records of patients visiting EDs across England between April 2017 and March 2018. These records were obtained from the HES.[26] HES contains routinely collected data covering procedures and treatments for all patients visiting secondary care facilities across England, tracking the patients' pathway from attendance to discharge. Attendance records contain an extensive range of information about patients, including the hospital site, admission and discharge dates and times, GP practice to which the patient is registered, and a comprehensive set of demographic information. HES also captures the patient's small geographical area of residence. This is at the lower layer super output area (LSOAs) level, which includes approximately 1500 individuals. These are used to link information on area-level of deprivation, namely the income component of the Index of Multiple Deprivation (IMD).[27]

Our target population is represented by patients aged 18–23 visiting EDs in England between April 2017 and March 2018, whose attendance was related to an episode of DSH (n=22 506). We focus on this period because of limitations in access to more recent data. Incidentally, this protects our analysis from any disruptions to services caused by the COVID-19 pandemic.

To obtain our analytical sample, we exclude attendances without a recorded LSOA (n=748). We also exclude records with missing values on key outcome or control variables (n=6349), and with implausible waiting time values (zero total attendance time or with waiting times recorded at the maximum value allowed by the system, n=507). To avoid ambiguity, we also exclude the few patients appearing more than once on the same day in the dataset (n=115). Furthermore, we exclude patients attending type 2 or type 3 EDs (n=713), hence focusing only on visits to type 1 EDs; that is 24-hour consultant-led services with full facilities for resuscitating patients. We do this as type 2 and type 3 EDs tend to have a different and less severe mix of patients.[28] Our final analytical sample is composed of 14 074 attendances categorised as related to episodes of DSH, across 127 EDs in England, between April 2017 and March 2018. In table 1, we show descriptive statistics of our main covariates for the entire sample and for the two subgroups defined by our exposure.

From the Office for National Statistics, we obtain data on the demographic structure of LSOAs. We use this to identify areas with a disproportionately higher share of individuals aged 18–25, an age-range consistent with being a university student. Online supplemental appendix 1 shows that the geographical distribution of these areas is aligned with the location of universities in England. From regular NHS Digital public releases,[29] we obtained the number of patients registered with GP practices divided in 5-year age groups. For each GP practice, we computed the share of young patients aged between 14 and 44.

### Exposure: proxy indicator for university students visiting EDs in England
Our main exposure is represented by a proxy indicator which aims to identify university students visiting an ED in England based on HES Accident and Emergency records. This proxy attempts to circumvent the absence of a specific variable indicating whether a patient is a

**Table 1** Descriptive statistics for attendances for the full sample and by exposure status

| | Full sample | | Not flagged as students | | Flagged as students | |
|---|---|---|---|---|---|---|
| **Variable** | **Perc.** | **Count** | **Perc.** | **Count** | **Perc.** | **Count** |
| Female | 64.65 | 9099 | 64.00 | 8357 | 73.03 | 742 |
| Male | 35.35 | 4975 | 36.00 | 4701 | 26.97 | 274 |
| Ethnicity | | | | | | |
| Any white background | 93.34 | 13 136 | 93.87 | 12 257 | 86.52 | 879 |
| Any black background | 1.45 | 204 | 1.33 | 174 | 2.95 | 30 |
| Any Indian background | 2.24 | 315 | 2.04 | 266 | 4.82 | 49 |
| Any mixed background | 2.98 | 419 | 2.76 | 361 | 5.71 | 58 |
| Living in rural area | 3.20 | 450 | 3.36 | 439 | 1.08 | 11 |
| Quintile of Index of Multiple Deprivation (income component) | | | | | | |
| Least deprived | 12.47 | 1755 | 11.26 | 1470 | 28.05 | 285 |
| Second quintile | 14.22 | 2002 | 14.28 | 1865 | 13.48 | 137 |
| Third quintile | 19.19 | 2701 | 19.02 | 2484 | 21.36 | 216 |
| Fourth quintile | 23.58 | 3319 | 23.66 | 3089 | 22.64 | 230 |
| Most deprived | 30.53 | 4297 | 31.78 | 4150 | 14.47 | 147 |
| ED arrival mode | | | | | | |
| Ambulance | 48.98 | 6893 | 49.14 | 6417 | 46.85 | 476 |
| Other | 50.67 | 7131 | 50.48 | 6592 | 53.05 | 539 |
| Not known | 0.36 | 50 | 0.38 | 49 | 0.10 | 1 |
| Referral source | | | | | | |
| Self-referred | 51.09 | 7191 | 51.01 | 6661 | 52.17 | 530 |
| GP | 1.58 | 222 | 1.53 | 200 | 2,17 | 22 |
| Emergency services | 31.04 | 4368 | 31.38 | 4097 | 26.67 | 271 |
| Healthcare providers | 8.50 | 1196 | 8.29 | 1083 | 11.12 | 113 |
| Work or education | 0.48 | 68 | 0.47 | 62 | 0.59 | 6 |
| Other | 7.31 | 1029 | 7.31 | 955 | 7.28 | 74 |
| Repeated DSH episodes | 35.00 | 4926 | 35.35 | 4616 | 30.51 | 310 |
| Out of hours (06:00–18:00) attendance | 58.18 | 8188 | 57.92 | 5495 | 61.52 | 625 |
| | **Mean (SD)** | **Range** | **Mean (SD)** | **Range** | **Mean (SD)** | **Range** |
| Daily ED attendances | 416.88 (184.94) | 88–1497 | 412.44 (184.16) | 88–1497 | 474.00 (185.61) | 145–1350 |
| Age | 20.28 (1.71) | 18–23 | 20.26 (1.72) | 18–23 | 20.43 (1.51) | 18–23 |
| Observations | 14 074 | | 13 058 | | 1016 | |

Notes: (1) The table shows percentages and counts for categorical or binary outcomes and mean/SD/range for discrete variables at the bottom. (2) The summary statistics refer to the full sample of deliberate self-harm (DSH) patients aged 18–23 (first column), and to the subsamples identified as non-students (second column) and students (third column) based on our student proxy described in the Methods section.
ED, emergency department.

student. First, we define 18–23 as our target age range consistent with most university students. Second, we identify LSOAs where more than 10% of residents are aged between 18 and 25. This second criterion tries to identify areas in England where student residences are located (see Appendix 1). Third, we identify GP practices whose patients are predominantly within an age range consistent with university students.[30] Specifically, we flagged 1016 DSH attendance from patients aged 18–23 living in LSOAs likely to host students and registered with GP practices with over 50% of patients aged 14–44 (hence likely to treat students). This is equivalent to 7.22% of our sample.

## Outcomes

Our analysis aims to study the healthcare provision and outcomes of university-aged students attending EDs for DSH, contrasting them with other DSH patients in the same age range. The outcomes studied cover the entire patient pathway in ED from arrival to discharge. However,

we do not focus on the choice of specific treatments delivered and their intensity as these are likely to depend on patient severity, which we are unable to measure. We first look at the time to initial assessment on arrival at the ED. Second, we look at the count of investigations delivered. Third, we look at various discharge destinations: patients refusing treatment or leave before being seen, referred to another provider or admitted to inpatient care, discharged with no follow-up. Finally, we measure unplanned follow-up visits within 7 days.

## Empirical approach

A simple comparison in means across groups of patients by student status is unlikely to give an accurate account of differences in outcomes due to various sources of confounding. For example, patients flagged as students may have more or less severe conditions than other DSH patients. They may also systematically visit EDs that are intrinsically different or at times where EDs are more or less busy or staffed. To study whether being a student is associated with differences in outcomes and treatments compared with other DHS patients visiting an English ED, we conduct a regression-based analysis. Our main empirical specification reads as follows:

$$y_{ihmd} = \beta \times Student_i + \gamma \times Ind_i + \omega \times Att_{imd}$$
$$+ \delta \; Severity_{ihmd} + HDFE_{hmd} + \epsilon_{ihmd} \quad (1)$$

In equation (1), $y_{ihmd}$ is the outcome observed for patient $i$ visiting ED $h$ on day $d$ of month $m$. The vector $Ind_i$ includes individual patient characteristics (age, ethnicity, whether lives in a rural area, income deprivation quintile of the LSOA of residence). The vector $Att_{imd}$ includes characteristics of the patient in relation to the ED attendance, namely: mode of arrival, referral source, whether it is a repeated DSH episode, whether the attendance happens in-hours (between 06:00 and 18:00) or out-of-hours (ie, between 18:00 and 06:00, during weekends or bank holidays[31]), and the volume of ED attendances on the same day. The volume of ED attendances on the same day accounts for differences in the level of pressure faced by ED staff, which may influence their behaviour towards patients. The primary ED diagnosis is used as a proxy for severity of DSH episode ($Severity_{ihmd}$). This concurs with ambulance arrivals to account for overall patient severity, which may impact how the DSH patient is treated in the ED. The list and frequencies of ED diagnoses is reported in Appendix 2. The vector HDFE represents a high-dimensional fixed effect that includes ED-specific seasonality combining month of year and day of week, in relation to the day of the ED attendance. The inclusion of HDFE allows us to account for various supply-side and demand-side observable and unobservable differences across EDs, and days of attendance, that may affect how DSH patients visiting an ED are treated.[31] In the absence of more detailed information of staffing at the time of the attendance, this allows to account for shifts in EDs, that are typically arranged well in advance to accommodate known weekly and seasonal fluctuations in ED demand.

Similarly, different EDs may have varied organisational structures and staff-skill mix; that is whether the staff are more or less at ease with patients visiting the department for an episode or DSH. Additionally, on the demand side, university students may be more likely to visits EDs for episodes of DSH at a specific time of year (eg, during end of term exam periods) or location (eg, hospitals located in major university cities).

Our main coefficient of interest is $\beta$ associated with the exposure variable $Student_i$. This reflects the difference in outcomes attributable to being a student according to our proxy definition. In order to interpret $\beta$ as an unbiased measure of our association of interest, we follow a standard selection on observables argument,[32] assuming conditional mean independence. In other words, conditional on the set of characteristics and the HDFEs we control for, the error term $\epsilon_{ihmd}$ is mean independent of our student dummy.

We estimate our models with two related approaches. First, for all outcomes, we report estimates of naïve models. These explain outcomes only with our exposure using a Probit model for binary outcomes and a Poisson model for right-skewed outcomes. We then compare our naïve estimates to results from our preferred empirical specification, with and without the more detailed controls for patient severity, represented by primary ED diagnosis. For binary models, we assume a linear probability model and estimate equation (1) using a feasible and efficient linear estimator, which accommodates HDFEs implemented in the Stata command *reghdfe*.[33] To account for the skewed distribution of waiting times and the number of investigations and procedures, we estimate models for these outcomes using a Poisson regression with a similar feasible efficient estimator designed to accommodate HDFEs implemented in the Stata command *ppmlhdfe*.[34] For Poisson models, we report results as average marginal effects.[35]

## Sensitivity analyses

To further explore the relationship between our student proxy and outcomes in ED for DSH attendances, we propose two stratified analyses. First, we estimate separately models for attendances during (1) standard working hours and (2) out-of-hours, respectively, as these periods may differ in the level of ED preparedness and resources. Second, we estimate two sets of models for (3) patients brought in by ambulance and (4) arrived at the ED by other means. The rationale for this is that patients arriving by ambulance are likely to have a higher priority. Finally, we estimate the models separately for patients visiting for a first or a repeated DSH episode during our study period. Unfortunately, we are not able to identify repeated DSH episodes before April 2017 due to data limitations. We estimate these sensitivity analyses using our preferred specification including control variables. We are unable to also control for primary ED diagnosis due to the emergence of singleton observations when

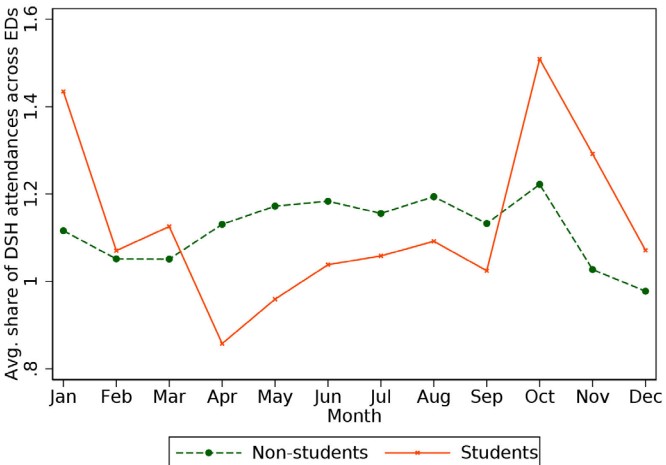

**Figure 1** Share of deliberate self-harm (DSH) patients of all emergency department (ED) attendances by month. Note: the figure shows the overall share of DSH attendances in each month across all 127 EDs in our sample, split between patients aged 18–23 identified as students and non-students using the proxy outlined in the Methods section.

partialling out the high-dimensional fixed when limiting sample size by splitting observations in subgroups.

## Patient and public involvement

This research did not involve members of the public or patients at any stage, beginning with developing the research questions to identifying the study design or outcomes. We did not consult members of the public in the interpretation of results, nor writing or editing the paper. We also have no specific plans to involve the public in the dissemination of the research findings. Nevertheless, Catherine Campbell developed this research while being enrolled as undergraduate student, that is, as a member of the main target group of this study. Similarly, Joe Dodd contributed to this research whilst being enrolled as a graduate student.

## RESULTS

### Seasonality of DSH ED attendances

As mentioned in the Methods section, there may be seasonality in the attendance of EDs for DSH patients. Figure 1 shows the seasonal variability in the share of patients attending EDs for an episode of DSH (averaged across the 127 EDs considered in our analysis), for patients identified as students and non-students.

The data suggest that approximately 1.1%–1.4% of all monthly ED attendances are related to DSH episodes. For patients identified as students, there is a clearer seasonality, with higher peaks in October and January, and lower prevalence compared with non-students during the summer months.

### Descriptive statistics

Table 1 shows characteristics of individuals in the sample, based on our main exposure. Patients aged 18–23 identified as students were more likely to be young women (73.03% compared with 63.00% for non-students) and less likely to have a white ethnic background (86.52% compared with 93.87% for non-students). Students were also less likely to live in rural and deprived areas. There are no substantial differences between students and non-students across the other covariates.

In table 2, we report summary statistics of outcomes by exposure status, with differences and corresponding test for difference in means and proportions. On average, patients aged 18–23 identified as students wait about 3.03 min longer for their initial assessment and receive 0.3 less investigations compared with non-students. They also seem to be less likely to be referred to another provider, to be admitted to hospital, and to have a follow-up visit scheduled at discharge. However, these descriptive differences may be confounded by various observed and unobserved differences between these two groups of young DSH patients.

**Table 2** Summary of outcomes by exposure status

| | Not flagged as students | | | | Flagged as students | | | | Difference | |
|---|---|---|---|---|---|---|---|---|---|---|
| Discrete outcomes | N | Mean | SD | Range | N | Mean | SD | Range | Diff. | t |
| Time to initial assessment | 13058 | 18.94 | 21.74 | 0–80 | 1016 | 21.97 | 22.74 | 0–80 | 3.03*** | 4.42 |
| Number of investigations | 13058 | 3.19 | 3.38 | 0–12 | 1016 | 2.89 | 3.40 | 0–12 | −0.29** | −2.67 |
| Binary outcomes | N | Perc. | Count | | N | Perc. | Count | | Diff. | z |
| Refused or left before treatment | 13058 | 6.03 | 787 | | 1016 | 5.81 | 59 | | −0.22 | −0.28 |
| Referred to another provider | 12509 | 42.06 | 5137 | | 978 | 34.25 | 335 | | −6.81*** | −4.17 |
| Admitted to hospital | 13058 | 31.92 | 4168 | | 1016 | 26.28 | 267 | | −5.64*** | −3.73 |
| Discharged with no follow-up | 12914 | 35.27 | 4555 | | 1010 | 41.09 | 415 | | 5.82*** | 3.72 |
| Unplanned follow-up within 7 days | 13058 | 6.46 | 844 | | 1016 | 4.92 | 50 | | −1.54 | −1.94 |

Notes: (1) the table shows percentages and counts for categorical or binary outcomes, and mean/SD/range for discrete outcomes at the top. (2) Third column includes the difference between students and non-students, alongside the corresponding t statistic testing the difference in means or z statistic testing the difference in proportions. Stars indicate statistical significance as follows: *p<0.05, **p<0.01, ***p<0.001.

**Table 3** Main results on full sample

| | | (1) | (2) | (3) |
|---|---|---|---|---|
| Time to initial assessment | Estimate | 3.032 | −0.958 | −1.033 |
| | 95% CI | (−1.207 to 7.271) | (−2.050 to 0.133) | (−2.130 to 0.0631) |
| | N | 14074 | 11581 | 11579 |
| Number of investigations | Estimate | −0.295 | −0.402** | −0.262* |
| | 95% CI | (−0.706 to 0.116) | (−0.680 to −0.125) | (−0.491 to −0.0327) |
| | N | 14074 | 11294 | 11289 |
| Refused or left before treatment | Estimate | −0.00220 | −0.0101 | −0.00914 |
| | 95% CI | (−0.0211 to 0.0167) | (−0.0308 to 0.0106) | (−0.0262 to 0.00790) |
| | N | 14074 | 11852 | 11852 |
| Referred to another provider | Estimate | −0.0681 | −0.0273 | −0.0107 |
| | 95% CI | (−0.143 to 0.00699) | (−0.0621 to 0.00753) | (−0.0529 to 0.0315) |
| | N | 13487 | 11261 | 11261 |
| Admitted to hospital | Estimate | −0.0564 | −0.0235 | −0.00143 |
| | 95% CI | (−0.118 to 0.00544) | (−0.0610 to 0.0140) | (−0.0437 to 0.0408) |
| | N | 14074 | 11852 | 11852 |
| Discharged with no follow-up | Estimate | 0.0582 | 0.00601 | −0.000976 |
| | 95% CI | (−0.0522 to 0.169) | (−0.0307 to 0.0427) | (−0.0408 to 0.0388) |
| | N | 13924 | 11715 | 11715 |
| Unplanned follow-up within 7 days | Estimate | −0.0154 | −0.0204 | −0.0221 |
| | 95% CI | (−0.0367 to 0.00584) | (−0.0440 to 0.00310) | (−0.0456 to 0.00145) |
| | N | 14074 | 11852 | 11852 |
| Controls | | No | Patient characteristics | Patient characteristics |
| | | | Daily att. volume at ED | Daily att. volume at ED |
| | | | | Primary ED diagnosis |

Notes: (a) all models and 3 include the following covariates: gender (female or male), age, ethnicity, whether lives in a rural area, income deprivation quintile of the layer super output area of residence, mode of arrival, referral source, whether it is a repeated deliberate self-harm episode, whether the attendance happens in-hours (between 06:00 and 18:00) or out-of-hours (between 18:00 and 06:00), the volume of emergency department (ED) attendances on the same day at the same ED, primary ED diagnosis. (b) Models were estimated with a Poisson feasible efficient estimator designed to accommodate high-dimensional fixed effects[34] for time to initial assessment and number of investigations. For all other outcomes, we estimated a linear probability model using a similar feasible efficient linear estimator accommodating high-dimensional fixed effects.[33] (c) Estimates represent average marginal effects for the difference between being flagged as students and non-student, 95% CIs in parentheses for SEs clustered at the ED level. Stars indicate statistical significance as follows: *p<0.05, **p<0.01, ***p<0.001.

**Main results**

Table 3 presents the estimated difference attributable to being a student for several indicators of ED performance, compared with non-students. We find a sizeable difference in the number of investigations delivered of −0.262 (−0.491 to −0.0327) during the visit (about 8% of the baseline average number of investigations for non-students). This difference is consistent across the three empirical specifications used (naïve model in column 1, adding patient characteristics and daily volume of attendances at the ED in column 2, and further accounting for differences in severity by controlling for the primary ED diagnosis in column 3). Our multivariate regression analysis suggests that all other differences emerging in the descriptive comparison in table 2 are likely confounded by other factors.

**Sensitivity analysis**

In table 4, we report estimates for our specification including control variables (but not primary ED diagnosis) for our three sensitivity checks, namely the stratified analyses by arrival time (in-hours vs out-of-hours), arrival mode (patients brought in by ambulance vs arriving at the ED by other means) and first or repeated DSH episode. Our main finding of fewer investigations delivered to patients identified as students seems concentrated among out-of-hours visits (column 2) and patients visiting for repeated DSH episodes (column 6, as opposed to the first DSH episode in the study period in column 5). Patient visiting during nights, weekends or bank holidays are also less likely to have an unplanned follow-up visit within 7 days (−0.0306 (−0.0576 to −0.00363)).

**Table 4** Stratified analyses

| | | | | Arrived by | Did not arrive | First DSH | Repeated DSH |
|---|---|---|---|---|---|---|---|
| | | In-hours | Out-of-hours | Ambulance | By ambulance | Episode | Episode |
| | | (1) | (2) | (3) | (4) | (5) | (6) |
| Time to initial assessment | Estimate | −1.511 | −0.820 | −1.997* | 1.611 | −1.007 | −1.653 |
| | 95% CI | (−3.875 to 0.852) | (−2.339 to 0.699) | (−3.703 to −0.292) | (−1.684 to 4.905) | (−2.754 to 0.739) | (−4.206 to 0.900) |
| | N | 3648 | 5758 | 4436 | 4734 | 6556 | 3013 |
| Number of investigations | Estimate | −0.387 | −0.485** | −0.545*** | −0.474* | −0.259 | −0.881** |
| | 95% CI | (−0.804 to 0.0295) | (−0.850 to −0.120) | (−0.839 to −0.250) | (−0.844 to −0.104) | (−0.577 to 0.0601) | (−1.510 to −0.252) |
| | N | 3581 | 5548 | 4533 | 4536 | 6386 | 2876 |
| Refused or left before treatment | Estimate | −0.00461 | −0.0157 | −0.0166 | −0.00294 | −0.0111 | −0.00441 |
| | 95% CI | (−0.0341 to 0.0248) | (−0.0433 to 0.0119) | (−0.0468 to 0.0135) | (−0.0298 to 0.0239) | (−0.0412 to 0.0191) | (−0.0360 to 0.0272) |
| | N | 3779 | 5910 | 4728 | 4973 | 6733 | 3125 |
| Referred to another provider | Estimate | −0.0334 | −0.0140 | −0.0708* | −0.0230 | −0.0202 | −0.0539 |
| | 95% CI | (−0.109 to 0.0425) | (−0.0585 to 0.0306) | (−0.140 to −0.00182) | (−0.0837 to 0.0377) | (−0.0633 to 0.0229) | (−0.169 to 0.0614) |
| | N | 3553 | 5603 | 4459 | 4707 | 6402 | 2922 |
| Admitted to hospital | Estimate | −0.00424 | −0.00631 | −0.0602 | −0.0119 | −0.0195 | 0.00206 |
| | 95% CI | (−0.0762 to 0.0678) | (−0.0476 to 0.0350) | (−0.127 to 0.00680) | (−0.0760 to 0.0522) | (−0.0534 to 0.0144) | (−0.0891 to 0.0932) |
| | N | 3779 | 5910 | 4728 | 4973 | 6733 | 3125 |
| Discharged with no follow-up | Estimate | −0.00424 | 0.00450 | 0.0276 | −0.0122 | 0.0260 | −0.0336 |
| | 95% CI | (−0.0762 to 0.0678) | (−0.0442 to 0.0532) | (−0.0258 to 0.0811) | (−0.0641 to 0.0397) | (−0.0156 to 0.0675) | (−0.117 to 0.0494) |
| | N | 3779 | 5829 | 4681 | 4885 | 6661 | 3070 |
| Unplanned follow-up within 7 days | Estimate | −0.00782 | −0.0306* | −0.0182 | −0.0211 | −0.0179 | −0.0442 |
| | 95% CI | (−0.0605 to 0.0449) | (−0.0576 to −0.00363) | (−0.0463 to 0.00997) | (−0.0533 to 0.0110) | (−0.0556 to 0.0199) | (−0.146 to 0.0573) |
| | N | 3728 | 5910 | 4728 | 4973 | 6733 | 3125 |

Notes: (a) all models and 3 include the following covariates: gender (female or male), age, ethnicity, whether lives in a rural area, income deprivation quintile of the layer super output area of residence, referral source, and the volume of emergency department (ED) attendances on the same day at the same ED. The variables used to stratify the sample (mode of arrival, whether it is a repeated deliberate self-harm episode, whether the attendance happens in-hours between 06:00 and 18:00 or out-of-hours between 18:00 and 06:00) have been included in the analyses not stratified by the relevant variable. (b) Models were estimated with a Poisson feasible efficient estimator designed to accommodate high-dimensional fixed effects[34] for time to initial assessment and number of investigations. For all other outcomes, we estimated a linear probability model using a similar feasible efficient linear estimator accommodating high-dimensional fixed effects.[33] (c) Estimates represent average marginal effects for the difference between being flagged as students and non-student, 95% CIs in parentheses for SEs clustered at the ED level. Stars indicate statistical significance as follows: *p<0.05, **p<0.01, ***p<0.001.

Columns 3 and 4 reflect the differences in outcomes attributable to whether the patient arrived by ambulance, comparing students and non-students. While the main results in table 3 account for the mode of arrival, splitting DSH patients by ambulance arrivals may help to better account for an intrinsic unobserved heterogeneity between these patient groups. We find that students who arrive by ambulance wait about two less minutes for initial assessment (−1.997 (−3.703 to −0.292)) and are about 7% less likely (−0.0708 (-0.140 to −0.00182)) to be referred to another provider. The number of investigations performed at a patient's initial assessment for both arrival modes is consistent with our main results.

## DISCUSSION
### Summary of findings
Our analysis proposes a method to identify patients who are likely students in the HES ED records, and explores differences in healthcare outcomes between students and non-students attending an ED after an episode of DSH. We find a statistically significant difference of 0.262 less investigations delivered to students compared with non-students, which is equivalent to about 8% compared with the baseline number of investigations for non-students. All other differences in ED care outcomes are not statistically significant in our main analysis. The results of our stratified analyses by time of attendance (regular times compared with out of hours, that is, 18:00 to 06:00 and weekends), ambulance arrival and first versus repeated DSH episode confirm the main findings providing a more nuanced picture. The reduced number of investigations (compared to non-students) is concentrated among students visiting EDs during out-of-hours and for repeated DSH episodes. Students visiting during out-of-hours are also less likely to have unplanned follow-up visits within 7 days. On the other hand, students arriving at the ED by ambulance seem to wait slightly less for initial assessment and be less likely to be referred to another provider.

### Strengths and weaknesses of the study
This paper has several strengths. First, our analysis uses a large dataset capturing ED attendances of DSH patients in English hospitals for an entire year. Second, it trials a student proxy indicator to identify and measure differences in emergency care outcomes compared with other patients attending EDs for the same reason—DSH. Third, in order to isolate the association between ED care outcomes and being a student, we address potential sources of confounding both on the demand (ie, patients) and supply (ie, hospital) sides. To account for compositional differences and unobserved patient severity, our main specification controls for patient characteristics and primary ED diagnosis. On the supply side, existing literature finds that—based on historical attendance patterns—some variation in attendance volume is predictable and as a result EDs are capable of adjusting capacity accordingly.[36] To circumvent the confounding

effect on estimates resulting from ED shift scheduling patterns and differing times of attendance, we control for number of ED attendances on the same day and account for an ED-specific month of year and day of week seasonality to clean our estimates from expected patterns of shift planning and natural variations in ED attendance across areas and seasons.

Despite our range of sensitivity checks, some limitations remain. First, our student indicator is only a proxy and should, therefore, be interpreted with caution. The marked difference in seasonality of DSH episodes (figure 1) and the geographical location of students' areas of residence (Appendix 1) provide some validation. Nevertheless, our method of identifying students is conservative, suggesting that our results are likely interpretable as lower bounds of student attendance volumes and related differences in ED care. Moreover, our study does not include any qualitative component; we do not interview ED staff members in regards to their attitude towards self-harm patients, a practice which would help to interpret some of our findings.[12] Finally, variability in the systems and protocol to assess and treat self-harm presentations may impede a meaningful comparison across areas.[37] We try to circumvent this latter aspect by accounting for hospital-specific characteristics (including ED-specific seasonality).

### Strengths and weaknesses in relation to other studies
A related study by Chartrand *et al*[38] proposes a longitudinal analysis of differences in discharge destination and rate of reattendance between patients who experience DSH and those who attempt suicide. They find those who reattended for DSH and suicidal attempt were more likely to be discharged to usual care during the initial assessment compared with a patient only attending for DSH. In view of this, a potential weakness of our study—dictated by data limitations—is that we did not include an indicator for suicide attempts. This is an important factor as patients who engage in DSH are at greater risk of subsequently dying by suicide.[16] Our results were consistent with a paper by Sinclair *et al*[39] whom employs a crossover methodological design to examine the differences in care before and after the introduction of a dedicated psychiatric nurse service. Importantly, the author finds an increase in the accuracy of assessment and diagnosis post intervention, wherein nurses are better trained to address DSH patients during initial assessments. These results may be consistent with a higher effectiveness in assessing patients, and hence explain the presented negative association between patient characteristics (being a student attending for DSH) and the number of investigations.

University students are not a homogenous group; rather there is large variation in health-seeking behaviours within the student population and compared with other patients in the same age group. Existing literature find that on arrival at the ED, students experience feelings of shame and fear surrounding a perceived 'failure to cope' and lack of confidentiality.[40 41] It is plausible that

these feelings are exacerbated when attending outside of normal working hours, pushing these patients to form a preference for not reattending the ED in case of need.[42]

The decreased number of investigations among students, compared with non-students, is consistent with physician's actions being guided by both a priority to treat physical symptoms (over psychological symptoms) and limited knowledge in terms of the immediate management of self-harm patients. In particular, qualitative literature finds that self-harm patients experience negative attitudes from staff and differences in treatment compared with other patients.[43] These patterns may originate from an increased level of uncertainty in dealing with this patient group, wherein a deficit in self-harm training has been emphasised.[44] Previous literature has also raised concerns over a lack of local policies that prioritise mental health over physical health.[45] Finally, the larger difference in the number of investigations for students visiting EDs for repeated DSH episodes may reflect either a better specific knowledge of the specific patient by ED staff or a marker of a generalised attitude towards students frequently attending EDs for mental health-related problems materialising in episodes of self-harm.[12 13]

In addition, the reduced number of investigations recorded among students compared with non-students irrespective of the arrival mode (ie, ambulance or not) may signal faults in the ED triage process. The first step on arrival requires patients to provide a brief summary of their health condition(s) at reception. Following this, patients wait for an initial assessment, wherein a medical or nursing professional determines priority for treatment by taking a brief medical history, a pain assessment and any collection of early warning scores.[46] Therefore, if staff are not trained in the management of mental health patients, EDs overlook a crucial opportunity to diagnose mental illness and commence a treatment pathway that may prevent subsequent self-harming behaviour.[47]

## Policy implications

Poor mental health is a major clinical priority given that it is at least partially preventable and a contributing risk factor for suicide attempts, especially among teens and young adults.[15] Although we do not identify worrying differences, our results may have implications for both university healthcare and ED systems. English universities need to further invest in resources to address the heightened demands for counselling services.[48] Oversubscription and a shortage of trained staff have impacted services through reduced length and number of sessions, longer waitlists and more referrals to off-campus providers.[2] In addition, university health centres need to prioritise information campaigns to decrease the stigma surrounding self-harm that prevents many students from seeking help, and consequently, lead to EDs being a first point of contact. The higher probability of refusing or leaving without treatment when attending EDs in-hours suggests the need for greater access to dedicated mental health staff or treatment, better access to community-based

resources, increased availability of inpatient psychiatric beds and a separate safe space in the ED.[49] In addition, roughly one half of DSH patients in ED do not receive a psychosocial assessment,[50] which further indicates the need to regulate and standardise mental health services. This suggest that better coordination between university mental health services and EDs may be needed in the long term. In the short term, in view of existing time and resource constraints, educating existing staff in the evaluation and management of DSH patients may be the only feasible approach to address such issues.

## Open questions and future research

This paper could be advanced in several ways. Further research is needed to investigate the association between students and ED outcomes among those who have current contact with mental health services at the time of self-harm. Future studies should aim to understand differences in student's health-seeking behaviour that could help design policies to mitigate staff bias towards this subgroup. Future research should also focus on the scarcity of integrated inpatient and outpatient mental health services at the county level. This is crucial in providing unified healthcare for a high risk and vulnerable population, while also reducing capacity constraints on EDs.

**Acknowledgements** We are grateful for comments and suggestions received from colleagues during Health Organisation, Policy and Economics (HOPE) seminars at the University of Manchester. Additional thanks to the Winter 2022 Health Economics Study Group (HESG) conference participants in Leeds for feedback during the poster session.

**Contributors** CC and IF developed the original idea for the study, designed the analysis and carried out the analysis. All authors contributed substantially to the interpretation of results and the writing of the manuscript. IF is responsible for the overall content as the guarantor.

**Funding** This research was supported by the National Institute for Health Research (NIHR) Applied Research Collaboration Greater Manchester (NIHR200174).

**Competing interests** None declared.

**Patient and public involvement** Patients and/or the public were not involved in the design, or conduct, or reporting, or dissemination plans of this research.

**Patient consent for publication** Not applicable.

**Provenance and peer review** Not commissioned; externally peer reviewed.

**Data availability statement** Data may be obtained from a third party and are not publicly available. The Hospital Episode Statistics data that support the findings of this study are available upon request from NHS Digital. Restrictions apply to the availability of these data. They were used under licence for this study only, and so are not publicly available.

**ORCID iD**
Igor Francetic http://orcid.org/0000-0002-2481-3749

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
