## [Reviewer comments · BMJ Open]

ARTICLE DETAILS

TITLE (PROVISIONAL)	Outcomes for university students following emergency care presentation for deliberate self-harm: a retrospective observational study of emergency departments in England for 2017/2018
AUTHORS	Campbell, Catherine; Dodd, Joe; Francetic, Igor

VERSION 1 – REVIEW

REVIEWER	Kronenberg, Christoph University Duisburg-Essen, CINCH
REVIEW RETURNED	02-Dec-2023

GENERAL COMMENTS	Referee report for: Outcomes for university students following emergency care presentation for deliberate self-harm: retrospective observational study of Emergency Departments in England for 2017/2018 (BMJ Open-2023-078672) The authors explore how to identify university students in UK hospital data and explore the differences in healthcare utilization after a hospital visit due to self-harm. The paper is well written and the analysis is conducted to a standard fitting of BMJ Open. I think there is a little room to polish the paper more for publication; see my comments below. Comment 1: The topic is relevant, and the relevance is adequately presented. However, there is some related literature on student mental health that does not get mentioned and I would have expected the research to be placed in that literature. Some examples: • Acampora, Michelle and Capozza, Francesco and Moghani, Vahid, Mental Health Literacy, Beliefs and Demand for Mental Health Support (October 29, 2022). Available at SSRN: https://ssrn.com/abstract=4261487 or http://dx.doi.org/10.2139/ssrn.4261487• Macchi, Elisa, Clara Sievert, Valentin Bolotnyy, and Paul Barreira. "Mental Health in European Economics Departments." (2023).• Bolotnyy, Valentin, Matthew Basilico, and Paul Barreira. 2022. "Graduate Student Mental Health: Lessons from American Economics Departments." Journal of Economic Literature, 60 (4): 1188-1222. Comment 2: While the presented way of identifying areas where students live, I am missing some validation. Does the list of areas identifying correspond to places that are close to universities?
---

	Minor Comment 1: The individual characteristics controlled for in the regression include people who live in a rural area. I don't imagine many students living in rural areas, so what is the justification for that variable? Minor comment 2: I find the definition of regular hours surprising. Do the results change with different definitions? Minor comment 3: What are implausible values on key variables? If you drop data for that reason, spell out what that means. Patients appearing multiple times a day could be a measure of severity. I would like to see whether the results change when they are included. Minor comment 4: Assuming that the authors use a user-written Stata/R package for HDFE, they should credit the package author with a citation. Minor comment 5: Many tables and figures do not have notes, which makes it hard to read them standalone.
--	--

REVIEWER	Poyraz Findik , Onur Istanbul Health and Technology University
REVIEW RETURNED	04-Dec-2023

GENERAL COMMENTS	The manuscript entitled "Outcomes for university students following emergency care presentation for deliberate self-harm: retrospective observational study of Emergency Departments in England for 2017/2018" has the aim to compare university-aged students and other patient groups attending emergency departments for deliberate self-harm in England. Thank you for the opportunity to review the manuscript. My suggestions are as follows:  1. Introduction: It is not suitable to start with the negative aspects of university on mental health. Mental health challenges tend to peak within the age group associated with university attendance. Concurrently, the university experience coincides with this critical period. Unfortunately, mental health issues can contribute to academic failure and even lead to school dropouts. Recognizing and addressing mental health concerns becomes crucial for these reasons. The introduction should be crafted in a manner that avoids portraying the university as a unique source of stress. For a considerable portion of individuals, the educational structure serves as a protective framework. 2. In the comparison between those who are students and those who are not, the difference in age groups (for instance, participants in the non-student group ranging from 10 to 60 years old) complicates the comparison. Here, there are variables other than just the impact of being a university student. It is unclear whether the comparison is focused on the effect of attending university or if it involves comparing this age group with other age groups, making it confusing. 3. The patients arriving outside regular hours (at night) or via an ambulance may be associated with more severe self-harming behavior. By making these inferences, connections can be established regarding the service delivery.
--

	4. Can longer waiting times for university students, leaving without being seen, and fewer examinations requested be associated with less severe self-harm behaviors? Can those continuing their education be considered to have lower psychopathology or be labeled as having an educational protective factor? 5. The addition of information such as those who have had previous mental health service contact, as well as individuals engaging in first-time or repeated self-harm attempts, to the data facilitates the interpretation of the results when presented the article. 6. The seasonal distribution of the student group, in terms of the impact of exam periods on self-harm, can be presented in separate graphs from the seasonal distribution of the non-student group. 7. It appears challenging for the article in its current state to reach a meaningful conclusion. The age distributions of the compared groups are quite different, making it difficult to understand the impact of being a student in this article structure. The groups should be reconsidered in a more suitable manner for comparison.
--	---

VERSION 1 – AUTHOR RESPONSE

Reviewer: 1

The authors explore how to identify university students in UK hospital data and explore the differences in healthcare utilization after a hospital visit due to self-harm. The paper is well written and the analysis is conducted to a standard fitting of BMJ Open. I think there is a little room to polish the paper more for publication; see my comments below.

Authors' response: Thank you for the generous assessment.

Comment 1: The topic is relevant, and the relevance is adequately presented. However, there is some related literature on student mental health that does not get mentioned and I would have expected the research to be placed in that literature. Some examples:

- Acampora, Michelle and Capozza, Francesco and Moghani, Vahid, Mental Health Literacy, Beliefs and Demand for Mental Health Support (October 29, 2022). Available at SSRN: <https://ssrn.com/abstract=4261487> or <http://dx.doi.org/10.2139/ssrn.4261487>
- Macchi, Elisa, Clara Sievert, Valentin Bolotnyy, and Paul Barreira. "Mental Health in European Economics Departments." (2023).
- Bolotnyy, Valentin, Matthew Basilico, and Paul Barreira. 2022. "Graduate Student Mental Health: Lessons from American Economics Departments." *Journal of Economic Literature*, 60 (4): 1188-1222.

Authors' response: Thank you very much for pointing us to his gap in the framing of the problem. We have now edited the introduction and discussion, incorporating the suggested papers. In the introduction, we now included the following paragraph:

"This paper builds on a growing body of literature regarding student mental health at university. Bolotnyy et al. [21] find that PhD economic students at top-ranked American universities have experienced depressive symptoms at a rate of over two times the population average, with only and a small percentage receiving help. Levels of depressive symptoms are even higher among graduate students attending European universities [22]. These studies demonstrate the prevalence

and severity of mental health in universities, common across countries. As mentioned, demand-side barriers such as misperceptions about care-seeking may contribute to the treatment gap among university students [23,24]. In light of this, Acampora et al. [25] finds that amongst individuals who receive information about the benefits of seeking help, females present a higher likelihood of seeking help, resulting in improved mental health scores at follow-up. Holt and Powell [15] find that 80 percent of university student attendees are not registered with a GP. This may reflect in an apparent misuse of emergency care services, which has been attributed to students failing to seek healthcare on campus due to fear of vulnerability and breached confidentiality [16].”

Comment 2: While the presented way of identifying areas where students live, I am missing some validation. Does the list of areas identifying correspond to places that are close to universities?

Authors’ response: Thanks for flagging this. We have now added some evidence validating our approach with Appendix 1 and Figure 1.

Minor Comment 1: The individual characteristics controlled for in the regression include people who live in a rural area. I don’t imagine many students living in rural areas, so what is the justification for that variable?

Authors’ response: Our rationale for including this variable is to adjust for differences between students and non-students. The latter are substantially more likely to live in rural areas, which may confound some of our outcomes. We hope this addresses the reviewer’s concern.

Minor comment 2: I find the definition of regular hours surprising. Do the results change with different definitions?

Authors’ response: Thanks for pointing this out. In the previous version of the manuscript the definition was incomplete. The correct definition of out-of-hours (and the one we use in our analysis) is “between 6PM and 6AM, during weekends or bank holidays”. This definition has been used in other studies of English EDs (e.g. <https://doi.org/10.1002/hec.4167>) as a good proxy for the out-of-hours times effectively adopted across the English NHS, in spite of some official documents using slightly different time limits (“6.30 pm to 8.00 am on weekdays and all day at weekends and on bank holidays” (see <https://www.nao.org.uk/wp-content/uploads/2014/09/Out-of-hours-GP-services-in-England1.pdf>)).

Minor comment 3: What are implausible values on key variables? If you drop data for that reason, spell out what that means. Patients appearing multiple times a day could be a measure of severity. I would like to see whether the results change when they are included.

Authors’ response: We have now provided further details in subsection “Data and sample”. The relevant paragraph now reads:

“To obtain our analytical sample, we exclude attendances without a recorded LSOA (n=748). We also exclude records with missing values on key outcome or control variables (n=6,349), and with implausible waiting time values (zero total attendance time or with waiting times recorded at the maximum value allowed by the system, n=507). To avoid ambiguity, we also exclude the few patients appearing more than once on the same day in the dataset (n=115). Furthermore, we exclude patients attending Type 2 or Type 3 EDs (n=713), hence focusing only on visits to Type 1 EDs; that is 24-hour consultant-led services with full facilities for resuscitating patients. We do this as Type 2 and Type 3 EDs tend to have a different and less severe mix of patients [29]. Our final analytical sample is composed by 14,074 attendances categorised as related to episodes of deliberate self-harm, across 127 EDs in England, between April 2017 and March 2018. In Table 1 below we show descriptive statistics of our main covariates for the entire sample and for the two subgroups defined by our exposure.”

Minor comment 4: Assuming that the authors use a user-written Stata/R package for HDFE, they should credit the package author with a citation.

Authors' response: Thanks for pointing this out. We have now adequately referenced the command used. This is reflected in the last paragraph of the "Empirical methods" section:

"We estimate our models with two related approaches. Firstly, for all outcomes we report estimates of naïve models. These explain outcomes only with our exposure using a Probit model for binary outcomes and a Poisson model for right-skewed outcomes. We then compare our naïve estimates to results from our preferred empirical specification, with and without the more detailed controls for patient severity, represented by primary ED diagnosis. For binary models we assume a linear probability model and estimate equation (1) using a feasible and efficient linear estimator, which accommodates high-dimensional fixed effects implemented in the Stata command `reghdfe` [34]. To account for the skewed distribution of waiting times and the number of investigations and procedures, we estimate models for these outcomes using a Poisson regression with a similar feasible efficient estimator designed to accommodate high-dimensional fixed effects implemented in the Stata command `ppmlhdfc` [35]. For Poisson models we report results as average marginal effects [36]."

Minor comment 5: Many tables and figures do not have notes, which makes it hard to read them stand-alone.

Authors' response: We have now completed all tables with notes to facilitate reading.

Reviewer: 2

The manuscript entitled "Outcomes for university students following emergency care presentation for deliberate self-harm: retrospective observational study of Emergency Departments in England for 2017/2018" has the aim to compare university-aged students and other patient groups attending emergency departments for deliberate self-harm in England.

Thank you for the opportunity to review the manuscript.

My suggestions are as follows:

1. Introduction: It is not suitable to start with the negative aspects of university on mental health. Mental health challenges tend to peak within the age group associated with university attendance. Concurrently, the university experience coincides with this critical period. Unfortunately, mental health issues can contribute to academic failure and even lead to school dropouts. Recognizing and addressing mental health concerns becomes crucial for these reasons. The introduction should be crafted in a manner that avoids portraying the university as a unique source of stress. For a considerable portion of individuals, the educational structure serves as a protective framework.

Authors' response: Thank you for highlighting this flaw in our motivating paragraphs. We have now edited the Introduction section accordingly.

2. In the comparison between those who are students and those who are not, the difference in age groups (for instance, participants in the non-student group ranging from 10 to 60 years old) complicates the comparison. Here, there are variables other than just the impact of being a university student. It is unclear whether the comparison is focused on the effect of attending university or if it involves comparing this age group with other age groups, making it confusing.

Authors' response: We initially meant to contrast ED outcomes for university students with all other patients visiting EDs after episodes of deliberate self-harm. Following up on this useful comment by the reviewer, we decided to refocus our analysis only to patients in the same age group. We are grateful for this suggestion, as this is a cleaner and more informative comparison. The manuscript has been edited to reflect this change, which we hope addresses the reviewer's concern. The relevant paragraph in the "Methods" section now reads as follows:

"Our target population is represented by patients aged 18 to 23 visiting EDs in England between April 2017 and March 2018, whose attendance was related to an episode of DSH (n=22,506). We focus on

this period because of limitations in access to more recent data. Incidentally, this protects our analysis from any disruptions to services caused by the COVID-19 pandemic.”

3. The patients arriving outside regular hours (at night) or via an ambulance may be associated with more severe self-harming behavior. By making these inferences, connections can be established regarding the service delivery.

4. Can longer waiting times for university students, leaving without being seen, and fewer examinations requested be associated with less severe self-harm behaviors? Can those continuing their education be considered to have lower psychopathology or be labeled as having an educational protective factor?

Authors’ response: Thank you for these suggestions. However, the conclusions are slightly different in our updated analysis focused on comparing students and non-students within a same age range (18-23). Accordingly, we feel like we cannot address these comments.

5. The addition of information such as those who have had previous mental health service contact, as well as individuals engaging in first-time or repeated self-harm attempts, to the data facilitates the interpretation of the results when presented the article.

Authors’ response: We agree with the reviewer that data on previous ED visits and mental healthcare use would be valuable. We partially followed the reviewer’s advice by proposing a stratified analysis splitting the sample between first and repeated deliberate self-harm episodes. Unfortunately, for this project we are limited to data on ED visit from March April 2017 to March 2018, meaning that we cannot link previous utilisation of other types of healthcare. We shall certainly consider this suggestion for future research.

6. The seasonal distribution of the student group, in terms of the impact of exam periods on self-harm, can be presented in separate graphs from the seasonal distribution of the non-student group.

Authors’ response: Thanks for this suggestion, which is now reflected in Figure 1.

7. It appears challenging for the article in its current state to reach a meaningful conclusion. The age distributions of the compared groups are quite different, making it difficult to understand the impact of being a student in this article structure. The groups should be reconsidered in a more suitable manner for comparison.

Authors’ response: We feel like this comment is addressed with the changes outlined in our response to point 2 above.

1

VERSION 2 – REVIEW

REVIEWER	Kronenberg, Christoph University Duisburg-Essen, CINCH
REVIEW RETURNED	09-Jan-2024

GENERAL COMMENTS	The edited version addresses all my minor concerns previously raised.
---

REVIEWER	Poyraz Findik , Onur Istanbul Health and Technology University
REVIEW RETURNED	16-Jan-2024

GENERAL COMMENTS	Thank you for completing the suggested revisions.
---